# Conservation and Targets of miR-71: A Systematic Review and Meta-Analysis

**DOI:** 10.3390/ncrna9040041

**Published:** 2023-07-26

**Authors:** Devin Naidoo, Ryan Brennan, Alexandre de Lencastre

**Affiliations:** 1Frank H. Netter MD School of Medicine, Quinnipiac University, Hamden, CT 06518, USA; 2Department of Molecular and Cellular Biology, Quinnipiac University, Hamden, CT 06518, USA

**Keywords:** miRNA, microRNA, miR-71, miRNA-71, longevity, aging, targets

## Abstract

MicroRNAs (miRNAs) perform a pivotal role in the regulation of gene expression across the animal kingdom. As negative regulators of gene expression, miRNAs have been shown to function in the genetic pathways that control many biological processes and have been implicated in roles in human disease. First identified as an aging-associated gene in *C. elegans*, miR-71, a miRNA, has a demonstrated capability of regulating processes in numerous different invertebrates, including platyhelminths, mollusks, and insects. In these organisms, miR-71 has been shown to affect a diverse range of pathways, including aging, development, and immune response. However, the exact mechanisms by which miR-71 regulates these pathways are not completely understood. In this paper, we review the identified functions of miR-71 across multiple organisms, including identified gene targets, pathways, and the conditions which affect regulatory action. Additionally, the degree of conservation of miR-71 in the evaluated organisms and the conservation of their predicted binding sites in target 3′ UTRs was measured. These studies may provide an insight on the patterns, interactions, and conditions in which miR-71 is able to exert genotypic and phenotypic influence.

## 1. Introduction

MicroRNAs (miRNAs) represent a relatively new class of RNAs that are among the most abundant gene regulatory molecules in eukaryotes [1]. The discovery of the first miRNA occurred when Rosalind Lee in the Ambros Lab discovered that *lin-4*, a gene that controls the larval development of *Caenorhabditis elegans*, produces a pair of small RNAs instead of encoding for a protein [2]. These *lin-4* RNAs were shown to have antisense complementarity to the 3′UTR (untranslated region) of the *lin-14* gene. Furthermore, mutations in *lin-14* rescued developmental abnormalities seen in *lin-4* loss-of-function mutations. These discoveries helped to develop a model in which the genetic interaction between a non-coding small RNA, *lin-4*, and a target gene, *lin-14*, is necessary for the transition from the first larval stage to the second larval stage of *C. elegans* development [3].

In 2005, Kim et al. discovered that miRNAs are transcribed by RNA polymerase II to form pri-miRNAs. Pri-miRNAs are then spliced, capped, and polyadenylated by similar mechanisms to how mRNAs are processed [4,5]. During transcription, the enzyme Drosha cleaves the end of the pri-miRNA to form precursor-microRNA (pre-miRNA). The pre-miRNA is then exported out of the nucleus by Exportin and is subsequently cleaved by the endonuclease Dicer to remove the stem loop and produce one mature miRNA (5′ arm) and one star strand (3′ arm) that is either degraded or has regulatory capabilities [6]. The mature miRNA is then loaded onto an Argonaute (AGO) protein to form the miRNA-induced silencing complex (RISC). The RISC complex containing Argonaute loaded with a miRNA targets the 3′ UTR of mRNAs that have complementarity to the bound miRNA. The “seed” region of the miRNA (nucleotides 2–7) has been shown to be the most important determinant of binding specificity. If the miRNA sequence binds with perfect or near perfect complementarity, the mRNA target is cleaved and destroyed. If the miRNA binds with some mismatches to the 3′ UTR of the target mRNA, translational repression occurs and the mRNA degrades later [7].

Since the discovery of the founding member of the miRNA family, *lin-4*, in *C. elegans*, 100–1000 s of other miRNAs have been identified, including *let-7*, which is conserved from *C. elegans* to humans [8]. At present, according to miRbase.org (accessed on 30 May 2023), 703 miRNAs in *C. elegans* and 4719 miRNAs have been discovered in humans [9]. An estimated 55% of miRNAs in *C. elegans* have homologs in humans [5]. Many miRNAs have been discovered to play roles in a number of different cellular and organismal pathways. For example, in humans, miR-155 has been shown to regulate macrophage defense against mycobacterium infection by increasing autophagy, inhibiting apoptosis, and regulating inflammation [10]. In addition, miR-124, expressed in *C. elegans*, has been shown to regulate a set of genes involved in ciliated neuron cell fate and function [3]. In *Echinococcus*, miR-10 regulates multiple signaling pathways, such as the MAPK and Wnt pathways [11]. Some miRNAs have been identified as tumor suppressors, such as let-7, which has the capability of negatively regulating a wide range of oncogenes involved in many different types of cancer [12]. Conversely, other miRNAs can act as oncogenes (oncomiRs), such as miR-24, which have been shown to negatively regulate tumor suppressor genes, such as MEN1, and contribute to cancer development [13]. These studies suggest that miRNAs can have multiple effects on different pathways and are important for regulating cellular and organismal function.

Original studies have shown that miRNA targeting is dependent on base complementarity between the 5′ end of a miRNA and the 3′ UTR of a target gene. However, recent studies have identified a fuller, more complex picture of miRNA/mRNA regulatory mechanisms. In regard to the relationship between miRNAs and target genes, studies have shown that one miRNA can regulate multiple genes, while one gene can be regulated by multiple miRNAs, depending on base complementarity [14]. Canonically, miRNAs function as negative regulators of gene expression through the RISC complex. However, a few exceptions have also been identified in the literature. MiRNAs have been shown to also promote the expression of genes instead of only downregulating through the RISC complex [15]. The proposed mechanism is through the miRNA recruitment of Argonaute and fragile X mental retardation-related protein 1 (FXR1) to form a micro-ribonucleoprotein (microRNP) for the up-regulation of translation [15,16]. However, this mechanism is unusual and an exception to the canonical role of miRNA downregulating expression. In addition, some miRNAs and their target genes participate in feedback systems wherein the miRNA negatively regulates a gene required for its production [17]. The image that emerges is that miRNAs and their targets form complex signaling pathways that underlie their complex roles in biology. This review aims to characterize conserved roles of a particular miRNA—miR-71.

The microRNA miR-71 was originally discovered in the pioneering cloning surveys in *C. elegans* [18]. Later, miR-71 was found to have dynamic changes in expression during development and adulthood and to be absolutely required for normal lifespan in *C. elegans* [19]. Since then, miR-71 has been found to be conserved in invertebrates and that it regulates multiple genetic pathways across species. Studies have shown that miR-71 can regulate several cellular processes, such as development, stress responses, and immune responses. In this systematic review, we investigate the role of miR-71 across invertebrates and the pathways and targets that it directly regulates. In addition, an analysis on the conservation of mature miR-71 sequences and target 3′ UTR sequences is performed. This review reveals the functional versatility of miR-71 across invertebrates and provides a bioinformatic framework to discover additional targets of miR-71.

## 2. Results

### 2.1. Targeted Pathways of miR-71

The targets of mir-71 in multiple organisms were organized by the pathway involved, which include Argonaute miRNA processing, developmental processes, environmental stress, pathogenicity and immunity, and insulin signaling (Table 1).

#### 2.1.1. Argonaute

Argonaute proteins are a component of miRNA function that bind to mature miRNAs guiding them to the mRNA targeted for either translation suppression or mRNA degradation. In humans there are only four functional Argonaute proteins (AGO1-4) which share similar structures but have slight differences [45]. Although the most robust and well-studied Argonaute protein is AGO2 the four Argonaute proteins have been found to share approximately 75% of their bound miRNA suite [45]. Studies have shown that the four Argonaute proteins can work redundantly to maintain homeostasis and compensate if one Argonaute protein is non-functional [46]. However when cells drastically change their environment a particular Argonaute protein loaded with a specific miRNA is required to execute specialized functions [46].

The association between *alg-1* an orthologue of *ago2* and miR-71 in *C. elegans* was validated by conducting an RNA pulldown and finding that miR-71 is pulled down with *alg-1* transcripts [21]. In *C. elegans* miR-71 has been shown to function in a negative feedback loop with the Argonaute protein genes *alg-1* and *alg-2* [17,22]. In miR-71 knockout mutants the levels of expression of *alg-1* and ALG-1 protein abundance are increased compared to the wild-type showing that miR-71 downregulates ALG-1 production [17]. Similarly the deletion of miR-71 binding sites on the *alg-*1 3′ UTR causes an increase in *alg-1* gene expression and ALG-1 protein abundance showing that miR-71 directly inhibits and regulates Argonaute protein expression [17]. In *C. elegans* specifically ALG-1 has been shown to be the predominant Argonaute protein for miRNA production and stability during larval development while ALG-2 is largely dispensable [23]. Given that *alg-1* is required for miRNA biosynthesis the proposed direct feedback regulatory loop between miR-71 and *alg-*1 establishes the homeostasis of miRNA levels [17]. In addition miR-71 has the possibility of also being associated with *alg-2* and may become more associated with *alg-2* as animals age [22]. This association with *alg-2* has been proposed to serve as a biological switch during aging given that *alg-2* mutation is associated with a shortened lifespan [22].

Indeed miR-71 interactions with *alg-1* and *alg-2* has systemic effects on global miRNA production seen by the increase in total miRNA production in miR-71 knockout mutants; this is believed to be due to the increased abundance of ALG-1 and ALG-2 Argonaute proteins. This change in miRNA production seen in miR-71 deletion mutants ultimately produces a dramatic dysregulation of total mRNA levels [17]. Additionally the mutation of *alg-1* produces an up-regulation in genes that have miR-71 binding sites for the 2–7 nucleotide seed sequence [23]. Implications on longevity have been found in the set of miRNAs regulated by *alg-1* [23]. These interactions are thought to work within the insulin/IGF-1 (IIS) pathway in a model that is dependent on miR-71 in association with *alg-1* [23]. Aalto and colleagues proposed that miR-71 ultimately represses the expression of *daf-2* within the IIS pathway allowing for an increased nuclear localization of *daf-16* which has been associated with increased longevity [23].

In mammals parasite-derived miR-71 can have effects on murine macrophages by targeting the mammalian Argonaute genes [20]. Parasites such as *E. multilocularis* have been shown to release miRNA-containing exosomes a type of extracellular vesicle to modulate their environment [20]. MiR-71 expressed from *E. multilocularis* infection can upregulate the Argonaute proteins AGO1 and AGO4 in mouse macrophages and ultimately lead to the repression of nitric oxide (NO) production [20]. This mechanism of miR-71 influencing macrophage Argonaute function in the parasite infection of mammals likely serves as a survival mechanism of the *E. multilocularis* organism to avoid the innate immune response [20]. Parasite-derived miR-71 transported in exosomes presents an interesting role of miR-71 in host–parasite interactions.

#### 2.1.2. Insulin Signaling

miR-71 was originally discovered to influence lifespan in *C. elegans* through interaction with the insulin signaling (IIS) pathway [19]. Mutants containing knockouts of miR-71 cause a drastic (50%) decrease in *C. elegans* lifespan while overexpressing miR-71 causes an increase in lifespan [19]. In addition miR-71 was shown to play an important role in stress resistance responses to heat and oxidative stress in *C. elegans* [19]. Older animals carrying miR-71 knockout mutations were shown to have an increased expression of PDK-1. PDK-1 is part of a signaling cascade in the IIS pathway that negatively regulates DAF-16 localization. As DAF-16 is a transcription factor required for normal longevity and stress resistance the up-regulation of PDK-1 in miR-71 knockout mutants explains the defective stress resistance and reduced lifespan seen in these mutants [19,47]. Although evidence suggests a role for miR-71 upstream of DAF-16 in the IIS pathway to promote longevity and an increased stress response *daf-16* mutations do not fully suppress the shortened lifespan phenotypes observed in miR-71 mutants suggesting that other gene pathways are regulated by miR-71 [19,42,47]. Indeed in the germline-less *glp-1*-mutant *C. elegans* miR-71 is required for the lifespan extension caused by germline removal [42]. One proposed target of miR-71 in the germline-mediated response is the gene TCER-1 which can also promote the nuclear localization of DAF-16 in the intestines [42]. In addition miR-71 has been shown to possibly play a role in the DNA damage checkpoint pathway through the inhibition of CDC-25.1 which is known to antagonize longevity [19].

In regards to the regulation of DAF-16 by miR-71 the expression of miR-71 in neurons was sufficient in upregulating DAF-16 nuclear translocation downstream in intestinal cells [42]. This model shows that the expression of miR-71 in neurons causes the cellular non-autonomous regulation of DAF-16 in the intestines. One proposed but untested model is that miR-71 might be transported from neurons to the intestines through extracellular vesicle non-autonomous regulation. Alternatively miR-71 regulates genes in neurons that trigger non-cell autonomous effects in intestinal DAF-16. Although neuronal miR-71 targets that affect stress resistance have been identified (which are discussed below) the mechanisms linking neuronal miR-71 to intestinal DAF-16 remain unknown.

The role of the IIS pathway in miR-71-mediated lifespan regulation extends to the response to food deprivation and starvation [43]. The role of the ISS pathway in miR-71-mediated regulation extends to the response to starvation which depends on the miR-71 regulation of AGE-1 a PI3K that is an upstream negative regulator of DAF-16 [43]. In addition miR-71 is predicted to target another gene *unc-31* which is also involved in the IIS pathway [43]. There is evidence that miR-71 can also regulate stress resistance and lifespan through regulating the genes *hbl-1* and *lin-42* in the developmental timing of *C. elegans* coming out of starvation [43].

Additionally another possible mechanism where miR-71 regulates longevity is through a pathway parallel to the IIS pathway [48]. After discovering that miR-71 does not directly regulate DAF-2 it was discovered that the capability of miR-71 in *C. elegans* to promote longevity does not solely depend on DAF-16 but at least partly acts in parallel to the IIS pathway [48]. There is evidence of another gene *kgb-1* which possibly interacted with miR-71 in the regulation of DAF-16 in the IIS pathway [44]. The activation of KGB-1 repressed the expression of miR-71 in *C. elegans* [44]. KGB-1 had previously been shown to antagonize the nuclear localization of DAF-16; however it was also shown that miR-71 could reciprocally regulate KGB-1 to promote DAF-16 nuclear localization [44].

#### 2.1.3. Development and Cell Signaling

miR-71 demonstrates diverse abilities pertaining to the signaling and developmental timing of many different organisms. Calcium signaling is essential for neuronal differentiation during larval development in *C. elegans* and the miR-71 targeting of the gene *tir-1* an adaptor protein in the calcium signaling pathway is required for asymmetric neuronal differentiation in the AWC olfactory neurons [30]. In *R. philippinarum* miR-71 affects shell color pigmentation by targeting the gene for calmodulin *CALM* [25]. In clams calcium signaling plays a key role in shell color patterning and the negative regulation of the calcium-signaling-pathway gene *CALM* contributes to shell color diversity among populations [25].

In addition to calcium signaling other signaling pathways required for development have been shown to depend on miR-71. It was discovered that miR-71 secreted from the parasite *S. Japonicum* has the capability of arresting the cell growth of hepatic tumor cells in mouse models [27]. In mammals the parasite-derived miR-71 targeting of the frizzled pathway gene *FZD4* inhibits hepatic cell signaling and can exert antitumor effects on hepatoma cells [27]. By negatively regulating the expression of the frizzled pathway protein FZD4 miR-71 effectively arrested hepatoma cells at the G0/G1 phase and inhibited the migration of tumor cells in mouse liver [27].

In *E. multilocularis* miR-71 is important in development by regulating several genes [28]. The regulation of these genes by miR-71 is required for the proper development of *E. multilocularis* as the inhibition of parasite early development was observed in miR-71 knockout tapeworms [28]. By knocking out miR-71 three genes with binding sites for miR-71 were all upregulated: frizzled serine/threonine kinase and T-cell immunomodulatory protein (EmTIP) [28]. Frizzled and serine/threonine kinase are involved in the signaling pathways for cell fate determination cell differentiation and cell migration [28]. Additionally EmTIP is important for cell proliferation and host–parasite communication [28]. The regulation of these genes by miR-71 remains imperative for cellular growth and early parasite development. In addition miR-71 can regulate asymmetric cell division in *E. multilocularis* by targeting nemo-like kinase (NLK) a protein in the signaling pathway for cell fate decisions [29]. The targeting of *nlk* by miR-71 is important in embryogenesis and development of the tapeworms by regulating cell fate and differentiation [29].

Targeting of the genes beta-14 xylanase and membrane metallo-endopeptidase-like 1 by miR-71 can affect metamorphic patterns in *R. venosa* [24]. The expression of miR-71 negatively regulates these two genes which have implications on *R. venosa* metamorphosis [24]. Throughout the larval stage as miR-71 expression decreases the expression of these genes increases leading to metamorphosis [24]. Additionally miR-71 can target chitin synthase (CHS1) a crucial enzyme for chitin biosynthesis to influence molting patterns in *L. migratoria* [26]. By increasing the expression of miR-71 decreased levels of CHS1 led to disrupted and abnormal molting patterns in locusts [26]. Conversely the knockout of miR-71 led to similar defects in molting in locusts showing a balancing modulation pattern between miR-71 and *CHS1* [26].

#### 2.1.4. Innate Immunity and Pathogenicity

In *M. japonicus* miR-71 targeted the calcification-associated peptide-1 (cap-1) to regulate viral infection and host autophagy in shrimps [38]. Autophagy is a critical component of the innate immune system in which the calculated degradation of intracellular compartments can be reallocated for other cellular utilities such as material recycling. Autophagy levels of *M. japonicus* were significantly increased in organisms overexpressing miR-71 and decreased in miR-71 knockouts. This can also work to the detriment of the host as organisms with elevated miR-71 had higher viral loads and mortality rates when infected with viruses that use autophagy as a part of their replicative process [38]. This evidence suggests that miR-71 can act as a bridge between viral infection and host autophagy [38].

In addition miR-71 and the miR-71-3p star sequence might be important for stimulating apoptosis while at the same time inhibiting the phagocytosis and phenoloxidase (an enzyme important in innate immunity) pathways in *M. japonicus* [49]. The down-regulation of miR-71 and miR-71-3p during apoptosis inhibition and the up-regulation of both during apoptosis activation hints at the role miR-71 and the miR-71-3p star sequence play in stimulating apoptosis [49]. Additionally the inhibition of phagocytosis and phenoloxidase activity showed an up-regulation of miR-71 and miR-71-3p showing that both sequences negatively regulate these pathways [49]. All three of these pathways are important for innate immunity in shrimp ultimately suggesting that miR-71 and the miR-71 star sequence can regulate the immune response [49]. However there were a lack of specific targets studied in these pathways by Yang et al. [49].

In *M. nipponense* the miR-71-3p star sequence has the ability to inhibit clotting through the targeting of *PCE* a pro-clotting enzyme [40]. The miR-71-3p star sequence is an important part of the immune response to bacterial infection of the river prawn through regulating clotting in conjunction with a long non-coding RNA (lncRNA) lncRNA transcript_11191 [40]. In response to *S. eriocheiris* infection miR-71-3p was upregulated and found to act in a trinity with lncRNA transcript_11191 and *PCE* [40]. The proposed mechanism from Ou et al. is of lncRNA transcript_11191 acting as a sponge and competitively binding to miR-71-3p to modulate the down-regulation of *PCE* [40].

Additionally miR-71 has been shown to target Seroin2 protein in the silkworm *B. mori* which is important for the antiviral immune response to alter infectivity [41]. Seroin proteins are silk-associated proteins in silkworms that protect the organism from microbes and infection [41]. By overexpressing miR-71 in *B. mori* there was a significant decrease in the expression of the Seroin2 protein showing that miR-71 can effectively regulate immune responses [41].

MiR-71 expressed in exosomes by the parasite *H. polygyrus bakeri* can also target mammalian interferon regulatory factor 4 (IRF4) in mouse lymphocytes and macrophages [39]. As previously described by Zheng et al. parasitic miRNAs can be utilized by mouse Argonaute protein for the targeting of mouse genes to regulate immune responses [20,39]. It has been shown that through this mechanism miR-71 has both anti-inflammatory and immunosuppressing properties by downregulating the expression of cytokines and chemokines in unpolarized mouse macrophages [39]. Another parasite *E. multilocularis* can also influence mouse macrophage function by ultimately repressing nitric oxide (NO) production an important substrate in the innate immune response [20]. These findings suggest a possible function of miR-71 to promote the infectivity of parasites in mammals by regulating the adaptive immune response through miRNA-containing exosome secretion and may serve as a potential target for therapeutic intervention [39].

#### 2.1.5. Environmental Stressors

Consistent with its function on longevity miR-71 has also been found to function on organismal response to various forms of environmental stresses. Dietary restriction is a conserved pro-longevity intervention that extends lifespan across species. In *C. elegans* dietary restriction has been shown to upregulate miR-71 leading to the repression of transcription factor *PHA-4* [36]. In turn the downregulating expression of *PHA-4* by miR-71 leads to an extended lifespan in *C. elegans* likely due to the increased expression of *DAF-16* and its target genes [36]. Additionally it has been shown that food odor can serve as a catalyst for the targeting of *tir-1* by miR-71 in *C. elegans* [31]. The olfaction-dependent targeting of *tir-1* by miR-71 promotes downstream effects such as proteostasis and longevity in these worms [31].

The development of the tapeworm *E. granulosus* relies on the miR-71 regulation of the oxidation reduction process through the targeting of the gene *HSC70* a heat-shock protein involved in maintaining proteostasis during oxidative stress [34]. During the normal development of tapeworms miR-71 levels increase throughout adulthood peaking at day 7 [34]. However the exposure of *E. granulosus* to bile salts in order to induce strobilation a method of asexual reproduction saw a decrease in the expression of miR-71 with a significant increase in the expression of the target gene *HSC70* [34].

Various stressors can also affect miR-71 interactions with transport proteins. In histone deacetylase (HDAC) knockouts a gene important for maintaining the stem cells of *S. mediterranea* the levels of miR-71 significantly decreased [35]. However upon exposure to radiation the downregulated miR-71 in HDAC knockout *S. mediterranea* was still capable of targeting the membrane transport protein MFS transporter DHA1 family solute carrier family 18 [35]. The regulation of this membrane transport protein by miR-71 may play a vital role during stem cell depletion in *S. mediterranea* [35].

Other stressors have shown more pronounced effects on miR-71 levels. Heavy metal exposure often induces effects by either serving as agonists or antagonists for proteins associated with signaling cascades or transport. In *D. pulex* cadmium affects the transport of cations such as calcium and copper [37]. Disruption can lead to the alteration of transporters and toxic accumulation of ions; therefore the organisms must have an adaptive response to acclimate to heavy metal exposure [37]. miR-71 expression is upregulated in cadmium-exposed *D. pulex* and targets the copper transporter *SLC31A1* to mitigate cadmium binding to the protein [37]. miR-71 also shows the capability of working synergistically with other miRNAs to regulate sodium calcium and potassium channels by targeting transporters such as *KCNN1 SCN2A* and *HCN2* [37].

An additional connection between sodium channels and miR-71 is also observed in *C. pipiens.* Pyrethroids such as deltamethrin have become widely used due to having potent insecticide properties with low human toxicity. One mechanism of action of pyrethroids is its inhibiting of the sodium inactivation channels in the nervous system. As a result insects exposed to pyrethroids experience uncontrolled depolarizations lack of repolarization prolonged contraction and twitches and eventually spastic paralysis before dying [50]. Enzymes such as Cytochrome p450 325BG3 can mitigate insecticide-induced damage by quickly degrading pyrethroids. miR-71 can target CYP325BG3 and silence the ability to degrade pyrethroids. Higher mortality rates of *C. pipiens* exposed to pyrethroids were observed in groups with enhanced miR-71 [32]. Meanwhile deltamethrin-resistant strains had a lower overall expression of miR-71 especially in females [33]. Evolution favors the selection of insects with decreased miR-71 expression leading to a decreased silencing of CYP325BG3 and thus an increase in the enzymes available to combat the insecticidal properties of pyrethroids such as deltamethrin.

### 2.2. Sequence Analysis

miR-71 is heavily conserved throughout invertebrates and has been reported to be expressed in 98 organisms (Appendix A). However no prior study has aligned the sequences to investigate what regions of miR-71 are the most conserved.

Sequences of the mature strand of miR-71 (miR-71-5p) from 106 studies were compiled and analyzed for conservation amongst different organisms. Strong conservation was seen in the seed sequence (nucleotides 2–7) and the anchor sequence (nucleotides 12–18) regions based on a threshold of 95% consensus with minimal conservation between these two sequences (Figure 1A). While nucleotides 2–7 are canonically referred to as the seed sequence important for target mRNA recognition a recent review on miRNA targeting has shown that nucleotides 13–16 provide additional interactions with the target once the seed region has been bound and are denoted the “anchor” sequence [51]. This is consistent with our findings that show that the seed sequence and anchor sequences are two of the most highly conserved regions of miR-71-5p. These regions have high conservation most likely due to the regulatory role they play in organisms. We found that the consensus 23 nucleotide sequence for the mature 5′ arm of miR-71 (miR-71-5p) generated by Clustal W is “UGAAAGACAUGGGUAGUGAGAUG” (Figure 1A).

However the star sequence (miR-71-3p) has less conservation throughout the sequence. The consensus 22 nucleotide sequence for the star strand 3′ arm of miR-71 generated by Clustal W is “UCUCACUACCUUGUCUUUCAUG” (Appendix A). Nucleotides 2 7–8 13–14 and 16–18 of miR-71-3p are conserved based on a threshold of 95% consensus showing a more scattered conservation pattern than miR-71-5p (Appendix A). Prior studies have suggested that the star sequences of miRNA have little targeting capacity and are mainly a degradation by-product from mature miRNA processing [52]. The scattered conservation of the miR-71 star strand may be explained by a lack of functional miRNA potential. This could suggest why there is little conservation seen in the star strand of miR-71 and why targets of miR-71-3p have not been heavily studied.

In addition we decided to conduct a sequence analysis of the proposed miR-71 targets from the 14 studies after secondary screening that published the predicted binding site sequences. The proposed target sites in the 3′ UTR were compiled and analyzed for conservation amongst the different invertebrate and mammalian genes targeted by miR-71. Heavy conservation was seen in the region of the 3′ UTR that is complementary to the seed sequence of miR-71 (Figure 1B). However little conservation is observed in the regions flanking the seed complementarity sequence in the 3′ UTR (Figure 1B). We believe that the conservation of this region of the 3′ UTR that is complementary to the seed sequence is meant for regulation by miR-71 and has been preserved in genes that are targeted by miR-71. However little conservation is seen in the region of the 3′ UTR complementary to the anchor sequence of miR-71.

Given the consensus sequences generated from the alignment of miR-71-5p and the proposed target 3′ UTRs we predict a consensus binding site for miR-71-5p consisting of an 8-mer A1 (adenosine complementary to nucleotide 1 of an miRNA) binding structure with the seed sequence and a heavy amount of binding with the anchor sequence (Figure 1C). The predicted 8-mer A1 binding structure represents the strongest binding capability of miRNAs to their target 3′ UTR with the adenosine positioned complementarily to nucleotide 1 of the miRNA believed to aid in Argonaute protein recognition [53]. In addition the low free energy of binding −21.9 kJ/mol presents an energetically favorable binding between these two consensus sequences (Figure 1C).

These results can be used in a bioinformatical approach to discover other possible targets of miR-71 based on these heavily conserved regions discovered in both the seed sequence of miR-71 and the complementary seed sequence of target 3′ UTRs.

## 3. Discussion

The most consistent evidence for the targets of miR-71 within invertebrates are the Argonaute system insulin-like signaling pathway and the toll-like receptor pathway (Figure 2).

In the Argonaute system miR-71 acts as a regulator of the ALG-1 and ALG-2 Argonaute proteins to control the synthesis of other miRNA production through a regulatory feedback loop. Argonaute proteins are required for miRNA and their mechanism of regulation through the stabilization of miRNA and target as well as recruiting the RISC complex for mRNA degradation. This negative regulation of Argonaute proteins by miR-71 regulates the abundance of global miRNA production and in turn their mRNA targets. This regulation is believed to lead to the homeostasis required for normal longevity and stress response.

miR-71 is also believed to affect the insulin-like signaling pathway through ultimate up-regulation of the nuclear transcription factor DAF-16 an orthologue of the human fox-head transcription factor FOXO. Prior evidence has shown that DAF-16 promotes increased longevity which seems to be stimulated through miR-71. However there are multiple possible gene targets in the IIS pathway that might be regulated by miR-71 to ultimately lead to the up-regulation of DAF-16. Many different models show that miR-71 can negatively regulate an inhibitor of DAF-16 expression to ultimately lead to DAF-16 up-regulation. This pathway may explain the longevity-promoting effects seen in organisms that overexpress miR-71.

The toll receptor domain TIR-1 pathway has also been shown to be regulated by miR-71 to promote proteostasis and inhibit neuronal degradation. TIR-1 is an orthologue of the human toll-like receptor adaptor protein SARM1 and has been shown to have deleterious effects in neuronal function and may promote neuronal degradation. In the neurons of invertebrates miR-71 negatively regulates TIR-1 and TIR-1 mutations suppress the defects of proteostasis response caused by the deletion of miR-71 knockout.

Interestingly miR-71 is also believed to play a role in the pathogenicity of parasitic invertebrates (Figure 3). Parasitic invertebrates excrete extracellular vesicles that have been shown to contain miR-71 and may influence the pathogenicity of parasites inside the host organism. miR-71 has been shown to upregulate Argonaute proteins within mammals which ultimately leads to a decrease in nitric oxide production by macrophages. This suggests that miR-71 can also act in upregulating target genes in a cell specific way that occurs outside of the host organism. In a separate pathway miR-71 secreted in extracellular vesicles has been shown to negatively regulate IRF4 and suppress the immune response in mammals by decreasing cytokine response and inflammation. It was shown that miR-71 can be loaded onto mammalian AGO2 to negatively regulate IRF4 and decrease the immune response. This most likely acts in a way that promotes the survival of the parasite within the host.

Overall the high conservation of miR-71 regulatory sequences (seed and anchor sequences) and the conservation of the 3′ UTR sequence that is complementary to the seed sequence can help to discover more possible target genes through a bioinformatic approach. These genes will need to be experimentally confirmed to assess whether they are indeed regulated by miR-71. With the knowledge that miR-71 has the capability of regulating multiple different genes future steps need to be taken to test whether there are other gene targets and pathways regulated by this miRNA and how they fit into systems biology.

## 4. Material and Methods

### 4.1. Search Strategy and Selection

A systematic search was performed using the PubMed Scopus and PubMed Central databases to identify studies that investigated miR-71 and its targets in invertebrates. The search was conducted from November 2022 to May 2023 and was limited to studies published in the English language. The search terms used were “microRNA-71” “miR-71” and “miR-71 targets”.

We independently screened the titles and abstracts of all identified studies to determine their eligibility for inclusion in the systematic review. The inclusion criteria were studies that: (1) experimentally investigated the gene targets of miR-71 (2) were primary research articles and (3) were written in the English language. The exclusion criteria were studies that: (1) did not experimentally investigate the protein targets of miR-71 (2) included vertebrates as the primary experimental subjects (3) were non-primary articles (systematic reviews and meta-analyses) or (4) were not written in English.

For the included studies we conducted a second independent screening to confirm study eligibility. A final selection following the Preferred Reporting Items for Systematic Reviews and Meta-Analyses (PRISMA) was conducted to show the filtering of databases and publications and the justification for exclusion throughout the screening process. From the search terms the three databases yielded 564 studies (63 PubMed 67 Scopus and 434 PubMed Central). Thirty-one studies remained after the primary and secondary screenings were completed (Figure 4).

### 4.2. Data Collection and Analysis

Data were collected from eligible studies by two independent reviewers and organized into a Microsoft Excel file (Appendix A). The extracted data included details such as species of invertebrate protein target identification general mechanism of action and a concise summary of the research as it pertains to miR-71.

After data collection the mechanisms of action were grouped into larger categories. Based on the findings the larger categories were determined to be the following: Argonaute System Insulin/Insulin-Like Signaling Pathway Development and Signaling Patterns Innate Immunity and Pathogenicity and Environmental Stress. For analysis human orthologues for prospective gene targets of miR-71 were discovered using RefSeq and the NCBI Gene Database.

### 4.3. Conservation of miR-71

Sequences for miR-71 strands were gathered after primary screening from a preliminary search of studies on three databases (PubMed Central PubMed and Scopus). A preliminary search showed 424 studies after removing duplicates. After primary screening 206 studies were further analyzed for inclusion and exclusion criteria as well as for sequences of miR-71. We compiled 112 sequences from the mature strand 5′ arm of miR-71 and 21 sequences from the star strand 3′ arm of miR-71 to align for sequence analysis (Appendix A). These sequences were compiled and analyzed by Clustal W to find a consensus sequence and regions of conservation. The sequence analysis conducted by Clustal W was then imported for viewing on JalView. The consensus sequence for miR-71-5p and consensus sequence for target 3′ UTR were finally analyzed for predicted binding using the RNAhybrid software.

### 4.4. Ethical Considerations

No ethical approval was required for this systematic review. All collected data are publicly available on the three search databases.

## Figures and Tables

**Figure 1 ncrna-09-00041-f001:**
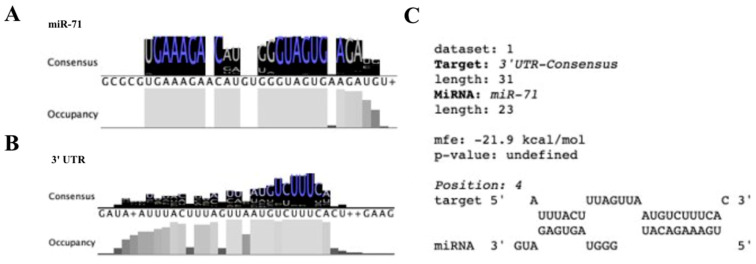
miR-71 and target 3′ UTR sequence conservation amongst multiple species. (**A**) A consensus sequence was created from the Clustal Omega alignment of 112 miR-71 sequences compiled after primary screening. Letters in blue indicate over 95% identity across sequences. (**B**) A consensus sequence was created from Clustal Omega alignment of 14 target 3′ UTR sequences after secondary screening. Letters in blue indicate over 95% identity across sequences. (**C**) Consensus sequences from (**A**,**B**) were aligned for miRNA-UTR binding showing an 8-mer A1 seed site the strongest binding capacity. Binding was favorable with a maximum free energy of −21.9 kcal/mol between the consensus miR-71 sequence and the consequence 3′ UTR sequence.

**Figure 2 ncrna-09-00041-f002:**
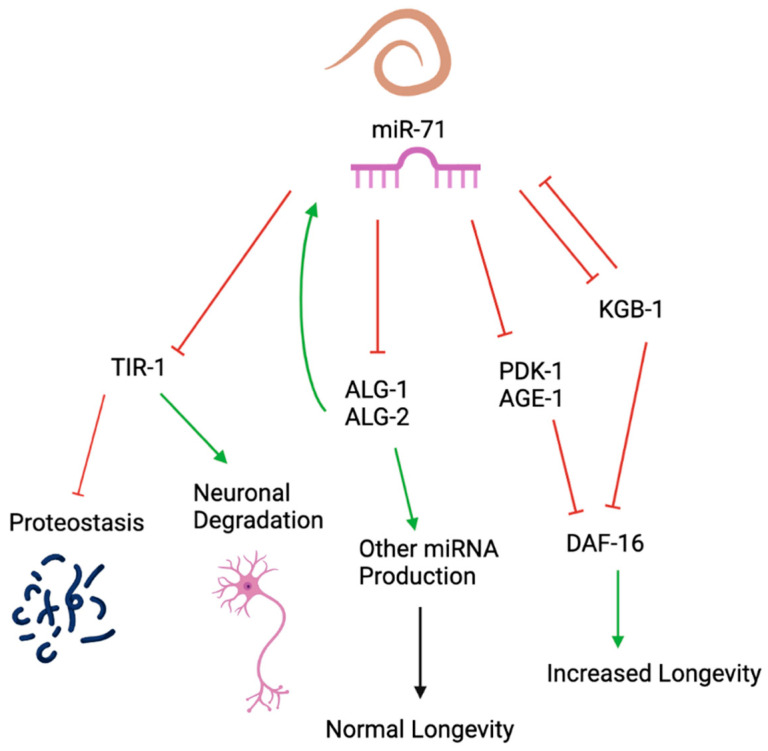
Major invertebrate targets of miR-71. Major pathways and genes regulated by miR-71 include Argonaute proteins insulin-like signaling (IIS) pathway and TIR-1.

**Figure 3 ncrna-09-00041-f003:**
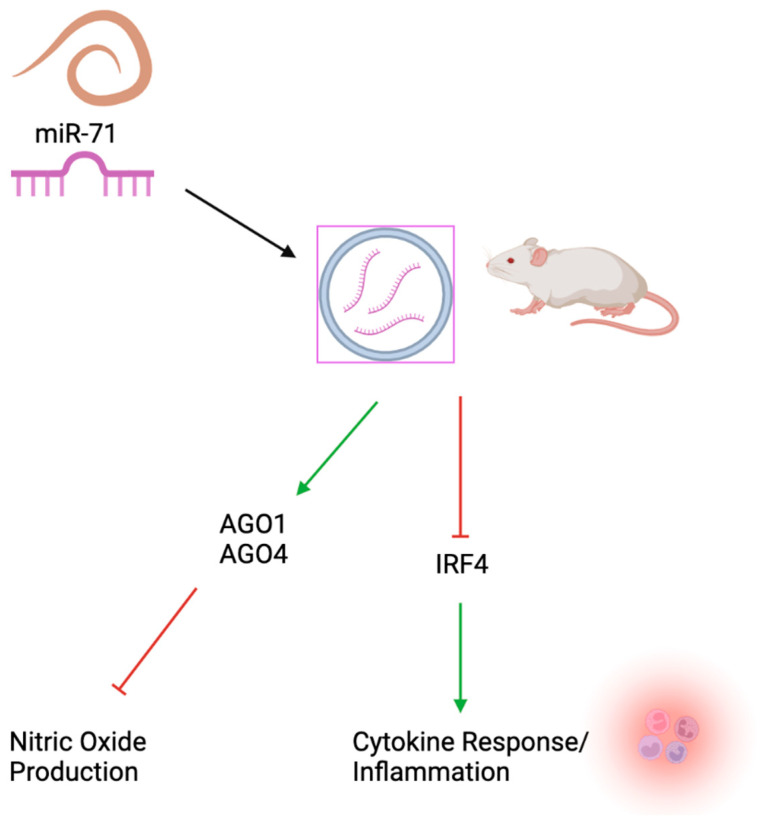
Mammalian targets of miR-71. miR-71 is believed to play a role in the pathogenicity of parasitic invertebrates. Parasitic invertebrates excrete extracellular vesicles that have been shown to contain miR-71 and may influence the pathogenicity of parasites. miR-71 has been shown to upregulate Argonaute proteins which ultimately leads to a decrease in nitric oxide production by macrophages. In a separate pathway miR-71 has been shown to negatively regulate IRF4 and suppress the immune response in mammals by decreasing cytokine response and inflammation.

**Figure 4 ncrna-09-00041-f004:**
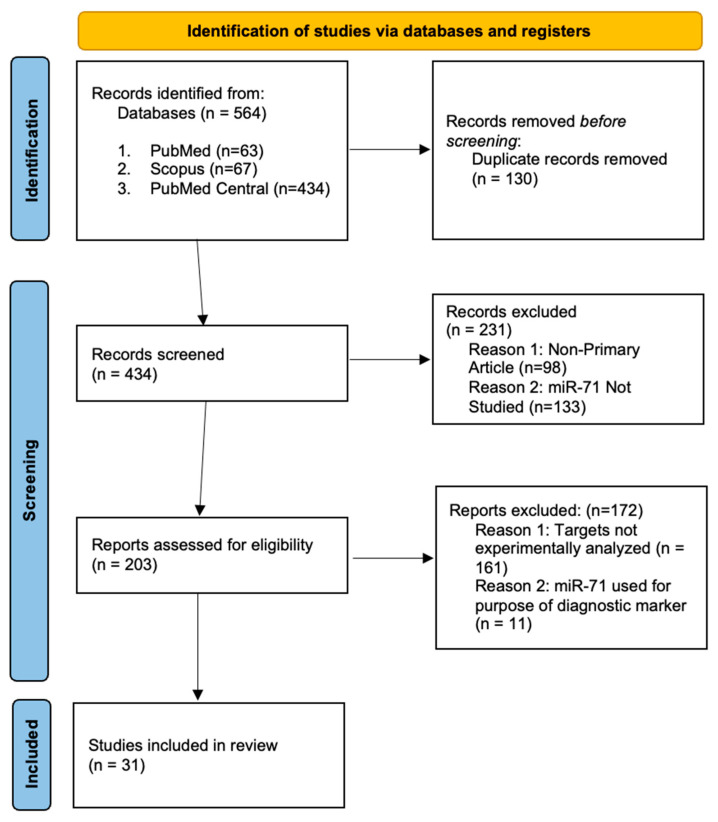
Preferred Reporting Items for Systematic Reviews and Meta-Analyses (PRISMA). A systematic search was performed using PubMed Scopus and PubMed Central databases to identify studies that experimentally investigate the targets of miR-71. The search terms used were “microRNA-71” “miR-71” and “miR-71 targets”.

**Table 1 ncrna-09-00041-t001:** Target genes of miR-71 and the pathways regulated. The targets of mir-71 from 31 studies after secondary screening were organized by the pathway involved. Human orthologues for each target gene were discovered using the HomoloGene database through NCBI.

miR-71 Target	Organism	Human Orthologue	Pathway	NLM Gene ID	Reference
AGO1	Mammals	AGO1	Argonaute System	2623	[20]
AGO4	Mammals	AGO4	Argonaute System	192670	[20]
ALG-1	*C. elegans*	AGO-2	Argonaute System	27161	[17,21,22,23]
ALG-2	*C. elegans*	AGO-2	Argonaute System	27161	[22]
C44F1.1	*C. elegans*	None	Argonaute System	N/A	[23]
Beta-14-Xylanase	*R. venosa*	None	Development and Signaling	N/A	[24]
Calm-1 (Calmodulin)	*R. philippinarum*	CALM1	Development and Signaling	801	[25]
Chitin Synthase	*L. migratoria*	None	Development and Signaling	N/A	[26]
FZD4 (Frizzled Pathway Protein)	Mammals	FZD4	Development and Signaling	8322	[27,28]
Membrane metallo-endopeptidase like 1	*R. venosa*	MMEL1	Development and Signaling	79258	[24]
Nemo-like Kinase	*E. multilocularis*	NLK	Development and Signaling	51701	[29]
Serine:Threonine Kinase	*E. multilocularis*	SNRK	Development and Signaling	54861	[28]
T Cell Immunomodulatory Protein	*E. multilocularis*	ITFG1	Development and Signaling	81533	[28]
TIR-1	*C. elegans*	SARM1	Development and Signaling	23098	[30,31]
Cytochrome P450 325BG3 (CYP325BG3)	*C. pipiens*	None	Environmental Stress	N/A	[32,33]
Heat Shock Cognate 70 kD protein (HSC70)	*E. granulosus*	HSPA4	Environmental Stress	3308	[34]
MFS transporter DHA1 family solute carrier family 18	*S. mediterranea*	SLC18A2	Environmental Stress	6571	[35]
PHA-4	*C. elegans*	FOXA1/FOXA2	Environmental Stress	3169/3170	[36]
SLC31A1 (Solute Carrier Family 31 Member 1)	*D. pulex*	SLC31A1	Environmental Stress	1317	[37]
Cap-1 (calcification-associated peptide)	*M. japonicus*	CAP1	Innate Immunity and Pathogenicity	10487	[38]
IRF4 (Interferon Regulatory Factor 4)	Mammals	IRF4	Innate Immunity and Pathogenicity	3662	[39]
PCE (Preclotting Enzyme)	*M. nipponense*	None	Innate Immunity and Pathogenicity	N/A	[40]
Seroin2	*B. mori*	None	Innate Immunity and Pathogenicity	N/A	[41]
CDC-25.1	*C. elegans*	CDC25A	Insulin Signaling	993	[19]
HBL-1	*C. elegans*	REST	Insulin Signaling	5978	[42]
TCER-1	*C. elegans*	TCERG1	Insulin Signaling	10915	[43]
KGB-1	*C. elegans*	MAPK8	Insulin Signaling	5598	[44]
lin-42	*C. elegans*	PER1/PER2	Insulin Signaling	5187	[43]
PDK-1	*C. elegans*	PDPK1	Insulin Signaling	8864	[19]
UNC-31	*C. elegans*	CADPS2	Insulin Signaling	93664	[43]

## Data Availability

Data are available upon request to the corresponding author.

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
