# Peer review of "Conservation and Targets of miR-71: A Systematic Review and Meta-Analysis"

_ncrna, 2023, doi:10.3390/ncrna9040041_

Round 1

Reviewer 1 Report

I compliment the authors for following a rigorous approach in preparing a systematic review of miR-71. However, minor modifications are:

1. Figure 2 should be Table 1.

2. The conclusion should be concise and should be based on the results presented. Authors should not add extra information in this section.

Author Response

  1. We have updated to table 1.
  2. Conclusion has been edited to better represent results.

Reviewer 2 Report

The authors give a systematic and comprehensive summary of conservation and the targets functions of mir-71, which contains a considerable wealth of information. The function of mir-71 is also described very clearly. But there are a few minor issues that need to be addressed before the article can be published.

1. The abstract emphasizes that the scope of the review is invertebrate, but the function of miRNA-71 in fish or mammals is also summarized at many places in the article. Please clarify the scope of the review.

2. There are two “Figures X” in Result 2.3, and the corresponding pictures are not found.

3. In this paper, mir-71-3p is referred to as star chain, which is an earlier way of calling, and it is suggested to change it to mir-71-3p.

4. The conclusion should be further refined. Some contents should be moved to the previous corresponding contents, such as the regulation of the Ago family and Toll receptors. Conclusions are best summarized with only critical, consistent information.

Author Response

  1. Abstract wording has been changed to broaden the scope.
  2. Figure X has been removed.
  3. We have adjusted wording to keep consistent with mir-71-3p.
  4. Conclusion has been better refined to represent results. 

Reviewer 3 Report

The systematic review on "Conservation and Targets of miR-71: A Systematic Review and Meta-Analysis" by Naidoo et al., would be an interesting addition in the area of miR-71 control of gene/protein expression. Overall, it is well-written but there are few comments to improve the manuscript. 

1. The abstract should include 1-2 sentences of results.

2. In introduction part, the citation style is not consistent, see the citation of Clarke et al., 2010. 

3. This systematic review is focused on miR-71 but the introduction part has little description of miR-71. More introduction of miR-71 is required in the introduction part. 

4. PRISMA flowchart has different numbers of studies as compared to mentioned in the text.

5. Figure 2 should be a table not figure. 

6. Divide the results part into results and discussion for better and easy understanding of main findings. 

7. Conclusion part is quite long so it can be merged in the discussion, and avoid citations in the conclusion. 

There are minor mistakes of language, so a careful editing is required.

Author Response

  1. Abstract has been updated to include results.
  2. Citations have been updated. 
  3. More information on miR-71 has been included in introduction. 
  4. PRISMA has been updated. 
  5. Figure has been changed to Table 1.
  6. Results and Conclusion have been edited. 
  7. Citations have been removed from the conclusion

Reviewer 4 Report

The Manuscript “Conservation and Targets of miR-71: A Systematic Review and Meta-Analysis”provides overview of the topic and objectives of the systemic review. It highlights the significance of microRNAs (miRNAs) in gene expression regulation and their association with various biological processes and human diseases. It specifically focuses on miR-71, which has been found to play a crucial role in diverse invertebrate organisms. The review aims to summarize the known functions of miR-71, including its target genes and pathways, while also examining its conservation across different species and the conservation of its predicted binding sites in target 3' UTRs. The findings from these studies have the potential to enhance our understanding of the regulatory mechanisms and the impact of miR-71 on genotypic and phenotypic traits.

 General Comments:

Overall, the manuscript effectively outlines the scope and purpose of the systemic review. It provides a clear context for the study, highlighting the importance of miRNAs and the specific focus on miR-71. However, there are a few areas that could be further improved for clarity and to enhance the overall impact of the abstract.

It would be valuable to emphasizing the unique aspects or contributions of this review. This could be in terms of new insights gained, methodologies employed, or potential implications for further research.

Since miR-71's functions are reviewed across multiple invertebrate organisms, it would be helpful to mention the specific species or taxonomic groups included in the analysis. This would provide clarity and context to the reader.

The text briefly mentions the compilation of sequences to quantify the conservation of miR-71 and its predicted binding sites. It would be beneficial to provide more information on the methodology used and any notable findings from this quantitative analysis.

By addressing these points, the Manuscript can be further refined to provide a clear and comprehensive overview of the systemic review, attracting the interest of readers and enhancing the understanding of the miR's functions and conservation in invertebrates.

Quality of english of upto the mark for publication.

Author Response

Edits have all been made to address the concerns of the reviewer.